# MaskINT: Video Editing via Interpolative Non-autoregressive Masked Transformers

## Abstract

Recent advances in generative AI have significantly enhanced image and video editing, particularly in the context of text prompt control. State-of-the-art approaches predominantly rely on diffusion models to accomplish these tasks. However, the computational demands of diffusion-based methods are substantial, often necessitating large-scale paired datasets for training, and therefore challenging the deployment in practical applications. This study addresses this challenge by breaking down the text-based video editing process into two stages. In the first stage, we leverage an existing text-to-image diffusion model to simultaneously edit a few selected key frames without any additional fine-tuning. In the second stage, we introduce an efficient model called MaskINT, which is built on non-autoregressive masked generative transformers. MaskINT specializes in frame interpolation between the key frames, benefiting from structural guidance provided by intermediate frames. The training of MaskINT incorporates masked token modeling. Our comprehensive set of experiments illustrates the efficacy and efficiency of MaskINT when compared to other diffusion-based methodologies. This research offers a practical solution for text-based video editing and showcases the potential of non-autoregressive masked generative transformers in this domain.

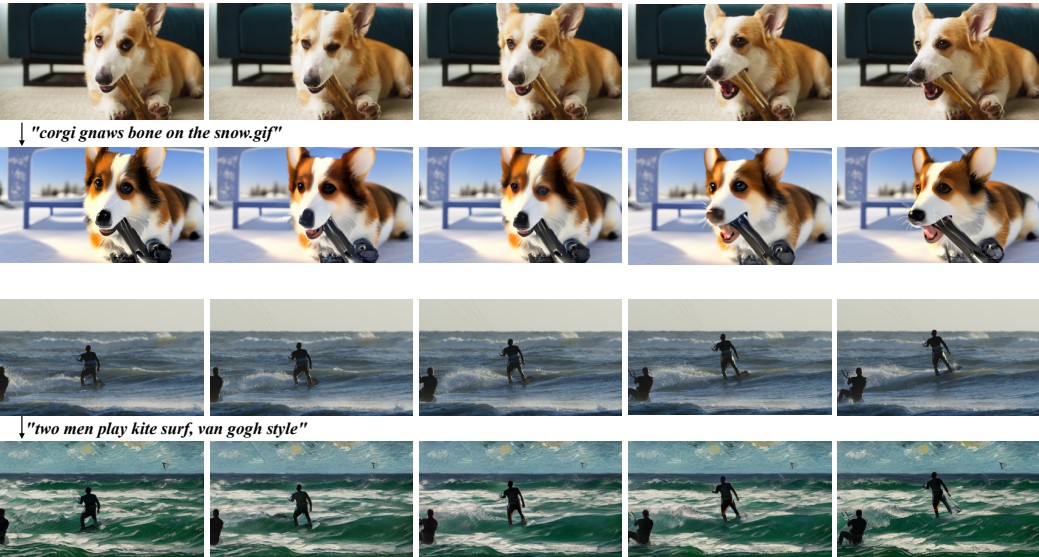

Figure 1: Examples of Video editing with MaskINT

## 1 Introduction

Text-based video editing, which aims to modify a video's content or style in accordance with a provided text description while preserving the motion and layout, plays a crucial role in a wide

range of applications, including advertisement, live streaming, and the movie industry, etc. This challenging task requires that edited video frames not only match the given text prompt but also ensure the consistency across all video frames.

Recently, numerous studies have showcased the impressive capabilities of diffusion models (Ho et al., 2020) in the domain of text-to-image and text-to-video generation (Rombach et al., 2022; Saharia et al., 2022; Blattmann et al., 2023; Singer et al., 2022). Later on, built upon Stable Diffusion (Rombach et al., 2022), several works (Zhang & Agrawala, 2023; Brooks et al., 2023; Tumanyan et al., 2023) have achieved remarkable success in the realm of text-based image editing. When extending to text-based video editing, existing works can be mainly divided into two ways: One is to train diffusion models with temporal modules on paired text-video datasets (Ho et al., 2022; Esser et al., 2023). However, due to the lack of extensive text-to-video datasets, these works typically struggle to achieve the same level of editing expertise seen in the real of image editing. Another approach involves leveraging a pre-trained text-to-image image diffusion models (i.e., stable diffusion (SD) ) for video editing task without additional training (Geyer et al., 2023; Qi et al., 2023; Khachatryan et al., 2023; Zhang et al., 2023a). These works achieve an overall consistency in appearance by extending the self-attention module in SD to perform cross-frame attention. Nevertheless, this attention-based temporal constraint remains implicit and suboptimal. Furthermore, while diffusion-based techniques are capable of producing high-fidelity videos, employing diffusion models to generate every single frame in a video is a laborious and redundant process. This renders it less viable for practical real-world applications.

In the meantime, studies indicate that non-autoregressive masked generative transformers (Chang et al., 2022; 2023; Yu et al., 2023; Gupta et al., 2023) can attain similar levels of performance in generating images or videos compared to diffusion-based methods. These works first employ VQGAN tokenizer (Esser et al., 2021) to encode image or videos into a sequence of discrete tokens, and then train transformers (Vaswani et al., 2017; Devlin et al., 2019) with masked token modeling. During the inference time, they employ non-autoregressive decoding, which generates all tokens in parallel and iteratively refine predictions in a few steps. Compared to diffusion-based models, masked generative transformers are usually significantly more efficient (Chang et al., 2023). Nonetheless, extending these techniques to perform global editing tasks, such as stylization, presents a formidable challenge. This type of task requires the replacement of nearly all tokens with new ones, as opposed to merely modifying tokens within a localized region.

In this paper, we disentangle text-based video editing into two separate stages. In the first stage, we utilize existing text-based image editing models (Zhang & Agrawala, 2023) to jointly edit a few key frames (i.e., the initial and last frames) from the video, guided by the provided text prompt. In the second stage, we propose a novel interpolative non-autoregressive generative transformers (MaskINT), which performs structure-aware frame interpolation by leveraging both the color information of the initial and final frames, as well as structural cues like edge maps from intermediate frames. Through disentanglement of key frame editing and frame interpolation into separate stages, our pipeline eliminates the requirement for paired video datasets during training, thereby enabling us to train the MaskINT in the second stage using video-only datasets. Furthermore, thanks to its non-autoregressive decoding, MaskINT significantly accelerates the generation of intermediate frames compared to using diffusion models for this purpose. We show that our method balances the trade off between quality and efficiency, offering comparable performance with existing diffusion methods yet taking much less time in generation.

Our major contributions are summarized as follows:

- We propose to disentangle the text-based video editing into a two stage pipeline, that involves key frames joint editing using existing image diffusion model and structure-aware frame interpolation with masked generative transformers trained on video only datasets.

- We propose MaskINT to perform structure-aware frame interpolation, which is the pioneer work that explicitly introduces structure control into masked generative transformers.

- Experimental results demonstrate that our method achieves comparable performance with diffusion methods in terms of temporal consistency and alignment with text prompts, while also providing 5-7 times faster inference times.

## 2 RELATED WORK

**Diffusion Models in Image Editing.** Leveraging the advancements of SD Rombach et al. (2022), numerous studies have achieved significant success in the field of text-based image editing Zhang & Agrawala (2023); Mou et al. (2023); Tumanyan et al. (2023); Kawar et al. (2023); Hertz et al. (2022); Zhang et al. (2023b); Parmar et al. (2023); Meng et al. (2021); Couairon et al. (2022). For example, ControlNet Zhang & Agrawala (2023), T2I-Adapter Mou et al. (2023), and Composer Huang et al. (2023) finetune SD with spatial condition such as depth maps and edge maps, enabling text-to-image synthesis with the same structure as the input image. The PNP Tumanyan et al. (2023) incorporates DDIM inversion features Song et al. (2020) from the input image into the text-to-image generation process alongside SD, enabling image editing without the necessity of additional training or fine-tuning. Instructpix2pix Brooks et al. (2023) trains a conditional diffusion model for text-guided image editing using synthetic paired examples, which avoid the tedious inversion. However, employing these methods on each video frame independently often leads to inconsistencies and flickering.

**Diffusion Models in Video Editing.** Recently, diffusion models also dominate the field of video generation and video editing Singer et al. (2022); Ho et al. (2022). For example, Gen-1 Esser et al. (2023) trains a video diffusion models on paired text-video datasets to generate videos with both depth map and text prompts. Meanwhile, several works utilize pre-trained image diffusion models to achieve video editing in a training-free way Wu et al. (2023); Yang et al. (2023); Qi et al. (2023); Khachatryan et al. (2023); Zhang et al. (2023a); Wang et al. (2023); Ceylan et al. (2023). To enable a cohesive global appearance among edited frames, a common approach in these studies involves extending the attention module of SD to encompass multiple frames and conducting cross-frame attention. In detail, Text2Video-Zero Khachatryan et al. (2023) performs cross-frame attention of each frame on the first frame to preserve appearance consistency. ControlVideo Zhang et al. (2023a) extends ControlNet with fully cross-frame attention to joint edit all frames and further improves the performance with interleaved-frame smoother. TokenFlow Geyer et al. (2023) enhance PNP Tumanyan et al. (2023) with extended-attention to jointly edit a few key frames at each denoising step and propagate them throughout the video based on nearest-neighbor field. On the contrary, we only utilize existing image diffusion models to edit two key frames, rather than all video frames.

**Generative Transformers.** Following GPT Brown et al. (2020), many pioneer works Esser et al. (2021); Yu et al. (2022); Le Moing et al. (2021); Ge et al. (2022); Hong et al. (2022) tokenize images/videos into discrete tokens, and train *Autoregressive Generative Transformers* to perform image/video generation, where tokens are generated sequentially based on previous output. However, these autoregressive methods become exceedingly time-consuming when the length of the token sequence increases. Recently, *Non-autoregressive Generative Transformers*, capable of simultaneously generating all tokens in parallel, have emerged as efficient solutions Chang et al. (2022); Yu et al. (2023). Specifically, MaskGiT Chang et al. (2022) first shows the capability and efficiency of this technique in image generation. It can be seamlessly extended to tasks like inpainting and extrapolation by applying various initial mask constraints. Muse Chang et al. (2023) achieves state-of-the-art performance in text-to-image generation by training on large-scale text-image datasets and brings significantly efficiency improvement. StyleDrop Sohn et al. (2023) further finetunes Muse with human feedback to perform text-to-image generation guided with a reference style image. Furthermore, MaskSketch Bashkirova et al. (2023) introduces implicit structural guidance into MaskGiT by calculating the similarity of attention maps in the sampling step. Nevertheless, this implicit structure condition is suboptimal.

In video generation, MaskViT Gupta et al. (2023) employ 2D tokenizer and trains a bidirectional window transformer to perform frame prediction. Phenaki Villegas et al. (2023) trains a masked transformer to generate short video clips condition on text prompt and extends it to arbitrary long video with different prompts in an autoregressive way. MAGVIT Yu et al. (2023) utilizes 3D tokenizer to quantize videos and trains a single model to perform multiple video generation tasks such as inpainting, outpainting, frame interpolation, etc. However, to the best of our knowledge, there is currently no existing literature in the field of text-based video editing utilizing masked generative transformers. Besides, there is a notable absence of research that delves into explicit structural control within this area.

**Video Frame Interpolation (VFI)** FI aims to generate intermediate images between a pair of frames, which can be applied to creating slow-motion videos and enhancing refresh rate. Advanced

methods typically entail estimating dense motions between frames, like optical flow, and subsequently warping the provided frames to generate intermediate ones Niklaus et al. (2017); Jiang et al. (2018); Sim et al. (2021); Lu et al. (2022); Reda et al. (2022); Huang et al. (2022). However, these methods are most effective with simple or monotonous motion. In our work, we perform frame interpolation by incorporating additional structural signals.

# 3 PRELIMINARIES

## 3.1 MASKED GENERATIVE TRANSFORMERS

Masked generative transformers (Chang et al., 2022) follow a two-stage pipeline. In the first stage, an image is quantized into a sequence of discrete tokens via a Vector-Quantized (VQ) auto-encoder (Esser et al., 2021). In detail, given an image $\mathbf{I} \in \mathbb{R}^{H \times W \times 3}$, an encoder $\mathcal{E}$ encodes it into a series of latent vectors and discretize them through a nearest neighbour look up in a codebook of quantized embeddings with size $M$. To this end, an image can be represented with a sequence of codebook's indices $\mathbf{Z} = [z_i]_{i=1}^{h \times w}, z_i \in \{1, 2, ..., M\}$, where $h$ and $w$ is the resolution of latent features. A decoder $\mathcal{D}$ can reconstruct the indices back to image $\mathcal{D}(\mathbf{Z}) \approx \mathbf{I}$. In the second stage, a bidirectional transformer model Vaswani et al. (2017) is learned with Masked Token Modeling (MTM). Specifically, during training, a random mask ratio $r \in (0, 1)$ is selected and $[\gamma(r) \cdot h \times w]$ tokens in $Z$ are replaced with a special [MASK] token, where $\gamma(r)$ is a mask scheduling function (Chang et al., 2022). We denote the corrupted sequence with masked tokens as $\bar{\mathbf{Z}}$ and conditions such as class labels or text description as $\mathbf{c}$. Given the training dataset $\mathbb{D}$, a BERT (Devlin et al., 2019) parameterized by $\Phi$ is learned to minimize the cross-entropy loss between the predicted and the ground truth token at each masked position:

$$\mathcal{L}_{MTM} = \mathbb{E}_{\mathbf{Z} \in \mathbb{D}} \left[ \sum_{z_i = [\text{MASK}]} - \log p_{\Phi}(z_i | \bar{\mathbf{Z}}, \mathbf{c}) \right] \tag{1}$$

During inference time, non-autoregressive decoding is applied to generate images. In detail, given the conditions $\mathbf{c}$, all tokens are initialized as [MASK] tokens. At step $k$, all tokens are predicted in parallel while only tokens with the highest prediction scores are kept. The remaining tokens with least prediction scores are masked out and repredicted in the next iteration. The mask ratio at each step is determined by $\gamma(\frac{k}{K})$, where $K$ is the total number of iteration steps.

## 3.2 CONTROLNET

**Latent Diffusion Models**   Denoising Diffusion Probabilistic Models (DDPM) (Ho et al., 2020) generate images through a progressive noise removal process applied to an initial Gaussian noisy image, carried out over a span of $T$ time steps. To enable efficient on high-resolution image generation, Latent Diffusion models (Rombach et al., 2022) (a.k.a. Stable Diffusion) operates the diffusion process in the latent space of an autoencoder. First, an encoder $\mathcal{E}$ compresses an image $\mathbf{I}$ to a low-resolution latent code $x = \mathcal{E}(\mathbf{I}) \in \mathbb{R}^{h \times w \times c}$. Second, a U-Net $\epsilon_{\theta}$ with attention modules (Vaswani et al., 2017) is trained to remove the noise with loss function:

$$\mathcal{L}_{LDM} = \mathbb{E}_{x_0, \epsilon \sim N(0,I), t \sim} \|\epsilon - \epsilon_{\theta}(x_t, t, \tau)\|_2^2, \tag{2}$$

where $\tau$ is the text prompt and $x_t$ is the noisy latent sample at timestep $t$. Stable Diffusion is trained on datasets with billion scale text-image pairs, which serves as the foundation model of many generation tasks.

**ControlNet**   In practice, it's challenging to design accurate text prompt to generate desired image. Furthermore, ControlNet (Zhang & Agrawala, 2023) is proposed to provide spatial layout conditions such as edge map, depth map, and human poses. In detail, ControlNet train the same U-Net architecture as SD and finetune it with specific conditions. We denote ControlNet as $\epsilon_{\theta}(x_t, t, \tau, s)$, where $s$ is the spatial layout condition.

# 4 METHODOLOGY

## 4.1 OVERVIEW

Figure 2 shows an overview of our framework. We disentangle the video editing task into key frame joint editing stage and structure-aware frame interpolation stage. Specifically, given a video clip

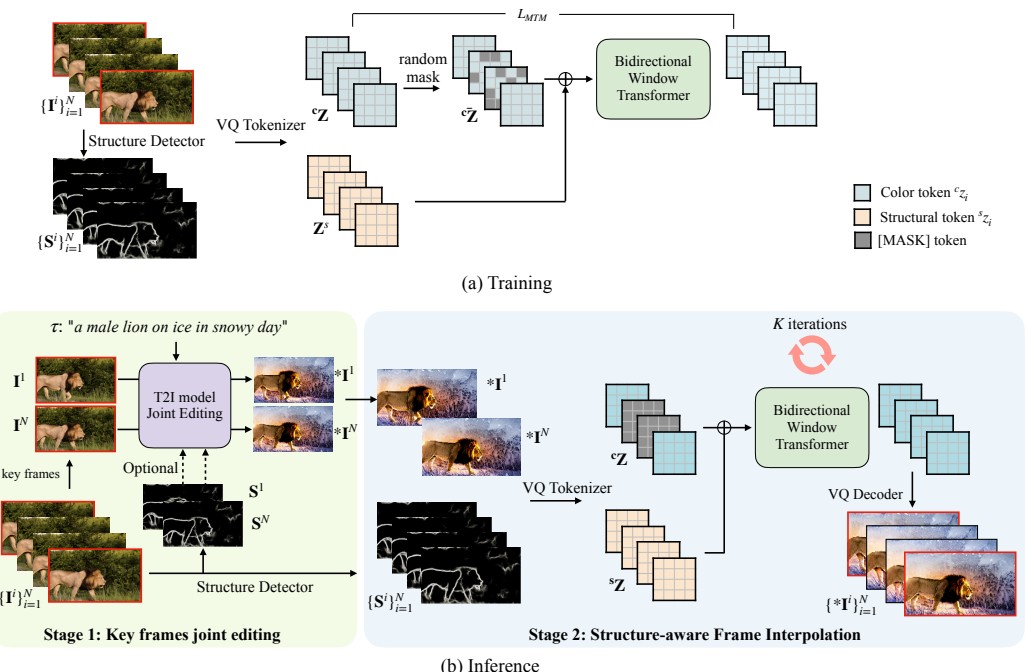

Figure 2: Overview of MaskINT. MaskINT disentangle the video editing task into two separate stages, i.e., key frame joint editing and structure-aware frame interpolation.

with $N$ frames $\{\mathbf{I}^i\}_{i=1}^N$, in the first stage, with the input text prompt $\tau$, we simultaneously edit two key frames, i.e., the initial frame $\mathbf{I}^1$ and last frame $\mathbf{I}^N$, using existing image-editing model $g(\cdot)$ (i.e., ControlNet (Zhang & Agrawala, 2023) ) that requires no additional tuning. This frame based joint editing module provides high-quality coherent edited frames $^*\mathbf{I}^1, {}^*\mathbf{I}^N = g(\mathbf{I}^1, \mathbf{I}^N, \tau)$ and is highly efficient on a pair of frame. In the second stage, MaskINT performs structure-aware frame interpolation via non-autoregressive transformers. The entire edited video frames are generated by $\{^*\mathbf{I}^i\}_{i=1}^N = f_\Phi(^*\mathbf{I}^1, {}^*\mathbf{I}^N, \{\mathbf{S}^i\}_{i=1}^N)$, where $\mathbf{S}^i \in [0,1]^{H \times W \times 1}$ is the structural condition (i.e., the HED edge map(Xie & Tu, 2015)). Our MaskINT is trained with masked token modeling (MTM) on video only datasets, conditioning on the structural signal $\{\mathbf{S}^i\}_{i=1}^N$ as well as two key frames (i.e., the initial frame $\mathbf{I}^1$ and last frame $\mathbf{I}^N$).

### 4.2 KEY FRAMES JOINT EDITING

To maintain the structure layout of selected key frames, we utilize ControlNet (Zhang & Agrawala, 2023) to edit $\mathbf{I}^1$ and $\mathbf{I}^N$ based on text prompt $\tau$ as well as their edge maps $\mathbf{S}^1$ and $\mathbf{S}^N$. However, even with the identical noise, applying ControlNet to each key frames individually, i.e., $\epsilon_\theta(x_t^1, t, \tau, \mathbf{S}^1)$ and $\epsilon_\theta(x_t^N, t, \tau, \mathbf{S}^N)$, cannot guarantee the appearance consistency.

To address this issue, following previous work (Wu et al., 2023; Geyer et al., 2023; Zhang et al., 2023a), we extend the self-attention blocks to simultaneously process multiple key frames. In the original U-Net of SD, the self-attention block projects the noisy feature map $x_t^j$ of $j^{th}$ frame at time step $t$ into query $\mathbf{Q}^j$, key $\mathbf{K}^j$, value $\mathbf{V}^j$. We extend the self-attention blocks to perform attention across all selected key frames by concatenating their keys and values and calculate the attention by

$$\text{JointAttn}(\mathbf{Q}^j, \{\mathbf{K}^j\}_{j \in \{1, N\}}, \{\mathbf{V}^j\}_{j \in \{1, N\}}) = \text{Softmax}(\frac{\mathbf{Q}^j[\mathbf{K}^1, \mathbf{K}^N]^{\mathbf{T}}}{\sqrt{c}})[\mathbf{V}^1, \mathbf{V}^N] \quad (3)$$

Note that although only two frames were used in Eq.3, this joint editing can seamlessly generalize to any number of key frames, at the cost of long processing time and huge demand in resources.

### 4.3 MASKINT

**Structure-aware Window-Restricted Transformer.** Previous masked generative transformers (Chang et al., 2022; Yu et al., 2023) employ a pure transformer with global attention (Vaswani et al., 2017). However, this architecture lacks the control of structure in generation, and it is computationally extensive when applied to videos. Thus, we propose the structure-aware window-restricted transformer, a simple yet effective method to integrating structure conditions into video generation. Specifically, we tokenize both RGB frames and edge maps with an off-the-shelf 2D VQ tokenizer (Esser et al., 2021). We utilize 2D VQ rather than 3D VQ (Yu et al., 2023) to accommodate varying numbers of frames and frame rate without constraints. We denote the tokens from RGB frames as $^{\mathbf{c}}\mathbf{Z} = \{^c z_i\}_{i=1}^{N \times h \times w}$ (color token) and tokens from edge maps as $^{\mathbf{s}}\mathbf{Z} = \{^s z_i\}_{i=1}^{N \times h \times w}$ (structure token), where $^s z_i, ^c z_i \in \{1, 2, ..., M\}$, where $M$ is the codebook size. Subsequently, two distinct embedding layers $e^C(\cdot)$ and $e^S(\cdot)$ are learned to map token indices $^{\mathbf{c}}\mathbf{Z}$ and $^{\mathbf{s}}\mathbf{Z}$ into their respective embedding spaces. Learnable 2D spatial positional encoding $\mathbf{P^S}$ and temporal positional encoding $\mathbf{P^T}$ are also added (Bertasius et al., 2021). Thus, the input can be formulated by $\mathbf{X} = e^c(^{\mathbf{c}}\mathbf{Z}) + e^s(^{\mathbf{s}}\mathbf{Z}) + \mathbf{P^S} + \mathbf{P^T} \in \mathbb{R}^{N \times h \times w \times c}$. Empirical results indicate that this straightforward addition operation of two types of embedding is remarkably effective to control the structure in generated frames. Moreover, given that there is no substantial motion between consecutive frames, we adopt self-attention within a restricted window to further mitigate computational overhead, following MaskViT (Gupta et al., 2023). In detail, our approach involves two distinct stages of attention. Initially, we employ spatial window attention, confining attention to tokens within a frame of dimensions $1 \times h \times w$. Subsequently, we extend this to spatial-temporal window attention, which confines attention to tokens within a tube of dimensions $N \times h_w \times w_w$, where $h_w$ and $w_w$ is the window size, where $h_w << h$ and $w_w << w$, which greatly reduce the complexity. Besides, to further reduce computation of transformer, we also add a shallow convolution layers to downsample $\mathbf{X}$ before the transformer encoder layers and an upsample layer at the end.

**Training.** By fully disentangling key frames editing and frame interpolation into distinct stages, our model no longer necessitates paired videos for training. Consequently, we can train MaskINT using video only datasets. Denote the color token of $i^{th}$ RGB frame as $^{\mathbf{c}}\mathbf{Z}^i = \{^c z^i\}_{i=1}^{h \times w}$. During the training time, we keep color tokens of the initial frame $^{\mathbf{c}}\mathbf{Z}^1$ and last frame $^{\mathbf{c}}\mathbf{Z}^N$, and randomly replace $[\gamma(r) \cdot (N - 2) \cdot N]$ color tokens of intermediate frames with the [MASK] tokens. We denote this corrupted video color tokens as $^{\mathbf{c}}\overline{\mathbf{Z}} = \{^{\mathbf{c}}\mathbf{Z}^1, ^{\mathbf{c}}\overline{\mathbf{Z}}^2, ..., ^{\mathbf{c}}\overline{\mathbf{Z}}^{N-1}, ^{\mathbf{c}}\mathbf{Z}^N\}$. The structure-aware window-restricted transformer with parameters $\Theta$ is trained by

$$\mathcal{L}_{MTM} = \mathop{\mathbb{E}}_{^{\mathbf{c}}\mathbf{Z}, ^{\mathbf{s}}\mathbf{Z} \in \mathbb{D}} \left[ \sum_{^c \overline{z}_i = [\text{MASK}]} - \log p_\Theta(z_i | ^{\mathbf{c}}\overline{\mathbf{Z}}, ^{\mathbf{s}}\mathbf{Z}) \right] \tag{4}$$

To avoid overfitting the structural condition, we random dropout a few structural tokens and replace them with zero embeddings during the training time.

**Inference.** During inference, our MaskINT can seamlessly generalize to perform frame interpolation between the jointly edited frames, although it is only trained with regular videos. Specifically, we tokenize the the initial and last edited frames $^*\mathbf{I}^1$ and $^*\mathbf{I}^N$ from Stage 1 into color tokens $^{\mathbf{c}}_*\mathbf{Z}^1$ and $^{\mathbf{c}}_*\mathbf{Z}^N$, and initialize color tokens of all intermediate frames $\{^{\mathbf{c}}\overline{\mathbf{Z}}^2, ..., ^{\mathbf{c}}\overline{\mathbf{Z}}^{N-1}\}$ with [MASK] tokens. We follow the iterative decoding in MaskGiT (Chang et al., 2022) with a total number of $K$ steps. At step $k$, we predict all color tokens in parallel and keep tokens with the highest confidence score.

## 5 EXPERIMENTS

### 5.1 SETTINGS

**Implementation Details.** We train our model with $100k$ videos from ShutterStock website[1]. During training time, we random select a $T = 16$ video clip with frame interval $1, 2, 4$ from each video and resize it to $384 \times 672$. We utilize Retina-VQ (Dubey et al., 2023) with 8 downsample ratio, i.e., each frame has $48 \times 84$ tokens. We employ Transformer-Base as our MaskINT and optimized it from scratch with the AdamW optimizer Loshchilov & Hutter (2019) for a duration of 100 epochs.

---

[1]https://www.shutterstock.com/video

The initial learning rate is set to $1e-4$ and decayed with cosine schedule. During the inference time, we set the number of decoding step $K$ to 32 and the temperature $t$ to 4.5.

**Evaluation.** Following (Wu et al., 2023), we use the 40 object-centric videos of the DAVIS dataset (Pont-Tuset et al., 2017), covering humans, animals, vehicles, etc. Besides, we also select 30 videos from the ShutterStock dataset. For each video, we manually design 5 edited prompts, including object editing, background changes and style transfers. Regarding the comparison methods, we select methods that built upon text-to-image diffusion models, including TokenFlow (Geyer et al., 2023), Text-to-video zero (Khachatryan et al., 2023), and ControlVideo (Zhang et al., 2023a). We also consider apply ControlNet to each frame individually with the same initial noise as baseline.

**Metrics.** Following previous works (Zhang et al., 2023a; Qi et al., 2023), we assess the quality of the generated videos using CLIP (Radford et al., 2021). In detail, we evaluate 1) temporal consistency (in short, Tem-Con), which calculates the average cosine similarity of all pairs of consecutive frames. 2) prompt consistency (in short, Tem-Con), which calculates the average cosine similarity between given text prompt and all video frames. To evaluate the efficiency, we report the duration required for generating a 16-frame video clip on a single NVIDIA A6000 GPU.

## 5.2 RESULTS

Table 1: Quantitative comparisons on the quality of the generated videos. "Tem-Con" stands for temporal consistency, and "Pro-Con" stands for prompt consistency.

| Method | DAVIS | | ShutterStock | | Time |
|---|---|---|---|---|---|
| | Tem-Con | Pro-Con | Tem-Con | Pro-Con | |
| ControlNet per frame (Zhang & Agrawala, 2023) | 0.9137 | 0.3136 | 0.9419 | 0.3040 | 50s |
| Text2Video-zero (Khachatryan et al., 2023) | 0.9642 | 0.3124 | 0.9811 | 0.3036 | 60s |
| TokenFlow (Geyer et al., 2023) | 0.9774 | 0.3169 | 0.9869 | 0.3133 | 150s |
| ControlVideo-edge (Zhang et al., 2023a) | 0.9746 | 0.3143 | 0.9864 | 0.3032 | 120s |
| MaskINT (ours) | 0.9519 | 0.3112 | 0.9714 | 0.3038 | 22s |

**Quantitative Comparisons.** Table 1 summarize the performance of these methods on both DAVIS and ShutterStock datasets. Notably, our method achieves comparable performance with diffusion methods, in terms of both temporal consistency and prompt consistency, while brings a significant acceleration in processing speed. In detail, MaskINT is almost 5.5 times faster than ControlVideo (Zhang et al., 2023a), whose fully cross-frame attention is computationally extensive. Moreover, MaskINT is nearly 7 times faster than TokenFlow (Geyer et al., 2023), whose DDIM inversion is time-consuming. On the contrary, our acceleration is derived from a combination of a relatively lightweight network design and a reduced number of decoding steps in masked generative transformers.

**Qualitative Comparisons.** Fig. 1(a) and Fig. 3 show samples of the edited videos with MaskINT. Our methods excel in producing temporally consistent videos that faithfully adhere to the provided text prompts. This extends to a wide range of applications, encompassing tasks such as stylization, background editing, foreground editing, and more. It also works well on challenging videos with substantial motion, such as dancing and running. Moreover, Fig. 4 provides a qualitative comparison of MaskINT to other baselines. Remarkably, diffusion methods (Geyer et al., 2023; Khachatryan et al., 2023; Zhang et al., 2023a) can ensure the consistency of overall appearance, but sometimes cannot maintain the consistency of detailed regions. For example, both TokenFlow (Geyer et al., 2023) and Text2Video-Zero(Khachatryan et al., 2023) exhibit noticeable artifacts in the leg region of the human subjects. ControlVideo (Zhang et al., 2023a) produces inconsistent hats. The potential explanation lies in the fact that these methods offer control over temporal consistency implicitly. Our MaskINT consistently interpolates the intermediate frames based on the structure condition and even maintain better consistency in local regions.

## 5.3 ABLATION STUDIES

**Number of of key frames** Although our model is trained with frame interpolation by default, MaskINT can seamlessly generalize to an arbitrary number of key frames. In this ablation study, we assess the impact of varying the quantity of key frames on performance on ShutterStock dataset. As shown in the left part of Table 2, with an increase in the number of key frames, the model

Input Frames                                        Generated Frames

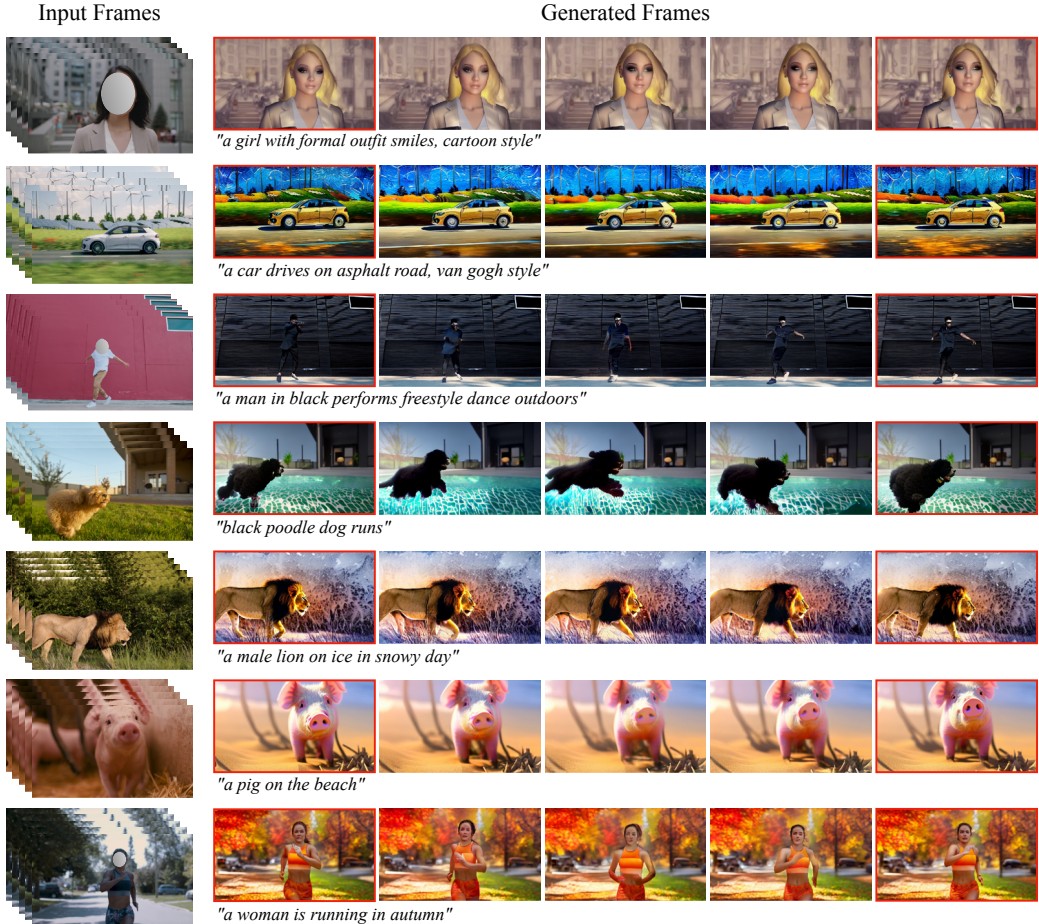

*"a girl with formal outfit smiles, cartoon style"*

*"a car drives on asphalt road, van gogh style"*

*"a man in black performs freestyle dance outdoors"*

*"black poodle dog runs"*

*"a male lion on ice in snowy day"*

*"a pig on the beach"*

*"a woman is running in autumn"*

Figure 3: Examples of video editing with MaskINT. Frames with red bounding box are jointly edited key frames. We mask out human faces in input video for privacy protection.

exhibits a marginal improvement in performance. Generally, with more information, performing frame interpolation is easier. However, simultaneously editing more key frames requires longer time due to the global attention among them.

0.9690 0.2984

Table 2: Ablation study on ShutterStock dataset of the number of key frames and the number of decoding steps $K$. "Tem-Con" stands for temporal consistency and "Pro-Con" stands for prompt consistency.

| # key frames | Tem-Con | Pro-Con | Time | decoding step $K$ | Tem-Con | Pro-Con | Time |
|---|---|---|---|---|---|---|---|
| 1 | 0.9690 | 0.2984 | 19s | 16 | 0.9691 | 0.3038 | 15s |
| 2 | 0.9714 | 0.3038 | 22s | 32 | 0.9714 | 0.3038 | 22s |
| 3 | 0.9721 | 0.3051 | 26s | 64 | 0.9719 | 0.3040 | 33s |
| 4 | 0.9728 | 0.3069 | 29s | 128 | 0.9720 | 0.3041 | 62s |
| 6 | 0.9737 | 0.3035 | 35s | | | | |

**Decoding steps** We also explore the number of decoding steps of the masked generative transformers in the second stage. The right part of Table 2 shows that more decoding steps can bring slight improvement on the temporal consistency. However, it tends to reach a saturation point and demands more computational time. Considering the trade-off between performance and efficiency, we chose $K = 32$ steps by default in all experiments.

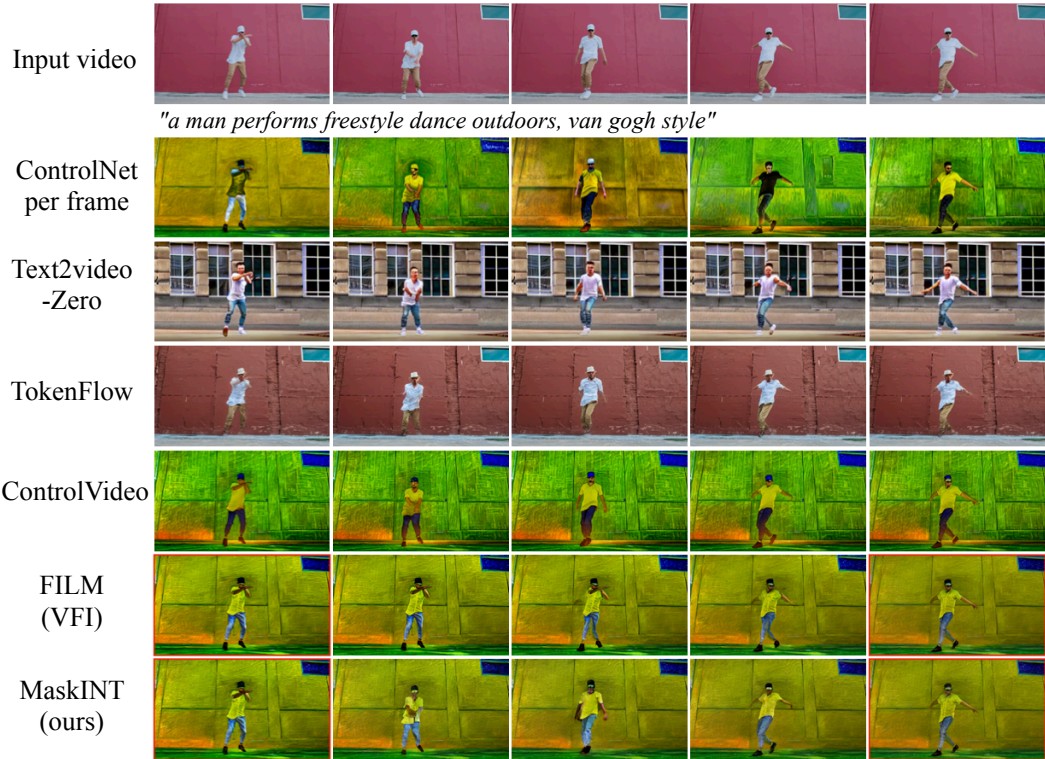

Figure 4: Qualitative comparisons with diffusion-based methods. More examples are shown in Appendix.

**key frames editing** We can also employ other controls to edit key frames in the first stage due to the disentanglement. In this study, we utilize ControlNet with depth map to perform key frame editing, and still use MaskINT trained with HED edge conditions. In this configuration, we achieve a "Tem-Con" score of $0.9713$ and a "Pro-Con" score of $0.3159$ on ShutterStock, a result that closely aligns with the performance of joint editing using HED edges. Fig. 5 shows an example of this setting. Editing key frames with depths provides more freedom over HED edge, but our method can still guarantee the consistency.

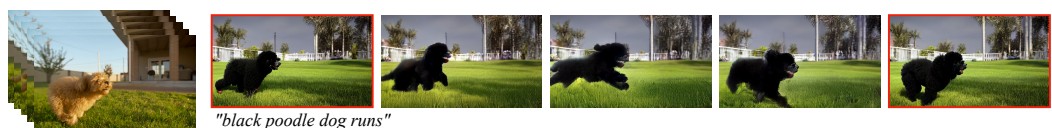

Figure 5: Examples of generated video from depth-based key frame editing.

## 6 LIMITATION AND FUTURE WORK

One potential limitation of our approach is that it still necessitates the use of diffusion-based methods for key frame editing in the initial stage, which can still be somewhat time-consuming. As for the future work, given that our approach disentangles video editing into two distinct stages, we intend to explore the integration of token-based methods, like Muse (Chang et al., 2023), for image editing. This endeavor aims to further enhance the efficiency of the initial stage.

## 7 CONCLUSION

In this paper, we propose MaskINT towards consistent and efficient video editing with text prompt. MaskINT disentangle this task into key frames joint editing with diffusion methods and structure-aware frame interpolation with non-autoregressive masked transformers. Quantitative and qualita-

tive experiments demonstrate that MaskINT achieves comparable performance with pure diffusion-based methods while significantly reduce the inference time. Our work demonstrates the substantial promise of non-autoregressive generative transformers within the realm of video editing.

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

## A    COMPARISONS WITH VIDEO FRAME INTERPOLATION (VFI)

We conduct a quantitative evaluation of the performance of MaskINT in the reconstructive setting, where it engages in video frame interpolation using the original key frames. In this evaluation, we compare the interpolated frames with the original video frames at the pixel level. We use same testing samples from DAVIS and Shutterstock datasets, employing peak signal-to-noise ratio (PSNR), learned perceptual image patch similarity (LPIPS), and structured similarity (SSIM) as the evaluation metrics. We benchmark our method against two state-of-the-art VFI methods, namely FILM Reda et al. (2022) and RIFE Huang et al. (2022). We also showcase the performance of applying VQ-GAN Esser et al. (2021) to all video frames, serving as an upper bound for our method.

As in Table 3, our method significantly outperforms VFI methods on all evaluation metrics, with the benefit of the structure guidance from the intermediate frames. Furthermore, Fig. 6 shows qualitative comparison between video frame interpolation methods FILM Reda et al. (2022) and MaskINT. Even when confronted with significant motion between two frames, our approach successfully reconstructs the original video, maintaining consistent motion through the aid of structural guidance. In contrast, FILM introduces undesirable artifacts, including disorted background, distorted cat hands, and the absence of a camel's head, etc. The major reason is that current VFI model mainly focus on generating slow-motion effects and enhancing frame rate, making them less effective in handling frames with large motions. Additionally, the absence of structural guidance poses a challenge for these methods in accurately aligning generated videos with the original motion.

Table 3: Quantitative comparisons on video frame interpolation with original keyframes.

| Method | DAVIS | | | ShutterStock | | |
|---|---|---|---|---|---|---|
| | PSNR↑ | SSIM↑ | LPIPS↓ | PSNR↑ | SSIM↑ | LPIPS↓ |
| RIFE Huang et al. (2022) | 17.31 | 0.5195 | 0.2512 | 20.44 | 0.7210 | 0.1533 |
| FILM Reda et al. (2022) | 17.00 | 0.5011 | 0.2363 | 20.90 | 0.7453 | 0.1246 |
| MaskINT | 22.15 | 0.6332 | 0.1483 | 24.19 | 0.7616 | 0.1097 |
| VQGAN per frame (ground truth) | 25.66 | 0.7429 | 0.0784 | 27.81 | 0.8327 | 0.0561 |

## B    LONG VIDEO GENERATION

Since the non-autoregressive pipeline generates all video frames simultaneously, it's challenging for it to edit an entire long video due to GPU memory limitation. However, our framework can still be extended to generate long videos by dividing the long video into short clips and progressively perform frame interpolation within each clip. Specifically, given a video with 60 frames, we select the 1st, 16th, 31st, 46th, and 60th frames as the key frames. We joint edit these 5 frames together and then perform structure-aware frame interpolation within each pair of consecutive key frames. As shown in Fig. 7, with this design, our method can still generate consistent long videos. Besides, in this proposed extension, the generation of later frames is decoupled from the generated early frames, which differs from the autoregressive long-video generation pipeline in Phenaki Villegas et al. (2023). Consequently, even if some early frames encounter difficulties, the generation of later frames can still proceed successfully.

## C    ADDITIONAL EXAMPLES FOR COMPARISONS.

We add more comparisons with diffusion methods in Fig. 8 and Fig. 9. Our methods can maintain the temporal consistency among variant examples.

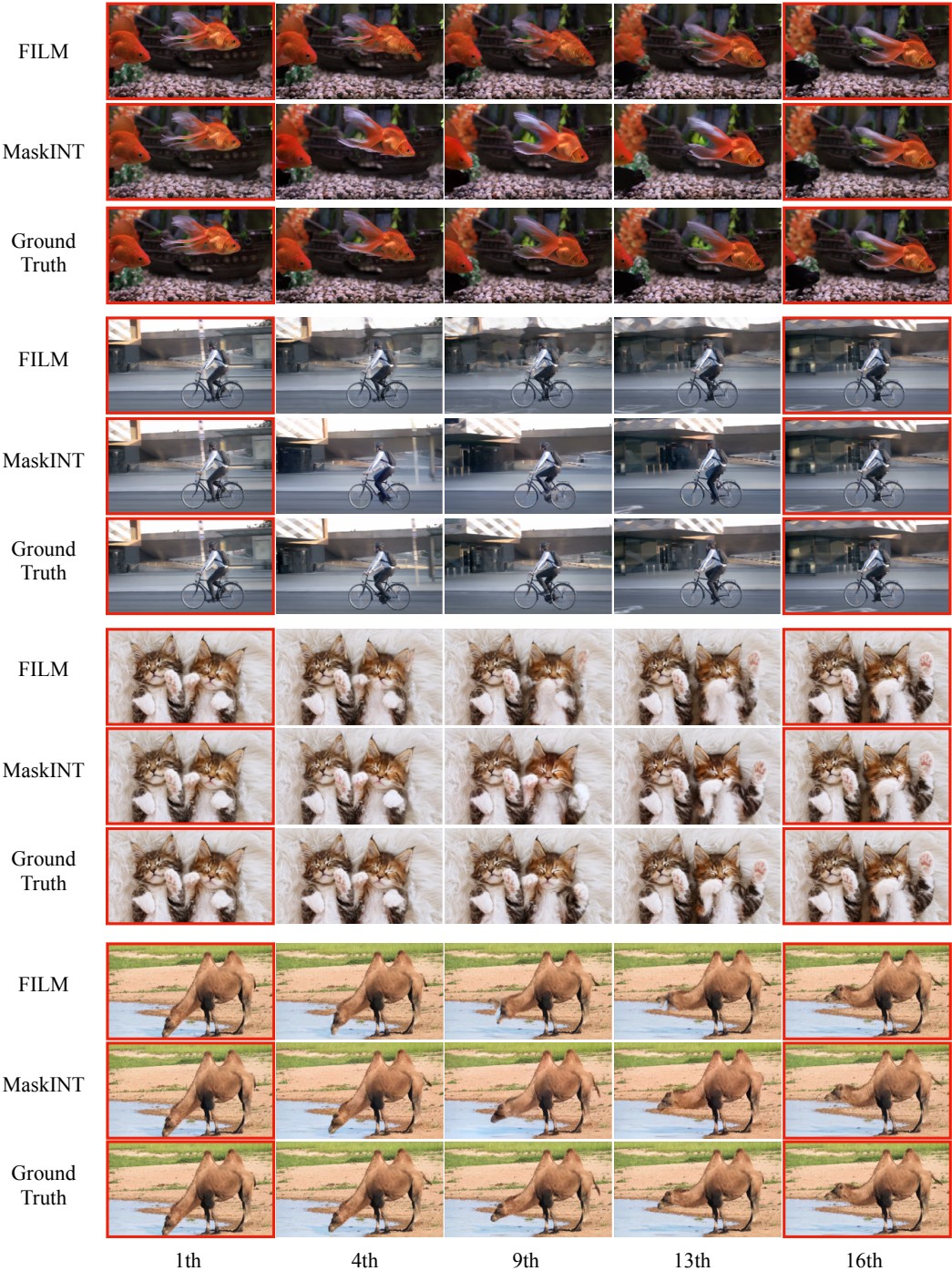

Figure 6: Qualitative comparisons between video frame interpolation FILM Reda et al. (2022) and MaskINT on video reconstruction task with original RGB frames. For each example, the 1st and the 16-th frames are given (images with red bounding box), and the intermediate 14 frames are generated. Here we show the 4th, 9th and 13th frames as an example. Please refer to the Supplementary for more examples and mp4 videos.

## D  ADDITIONAL EDITING EXAMPLES

We present more video editing examples in Figure 10 to show the generalization of our method.

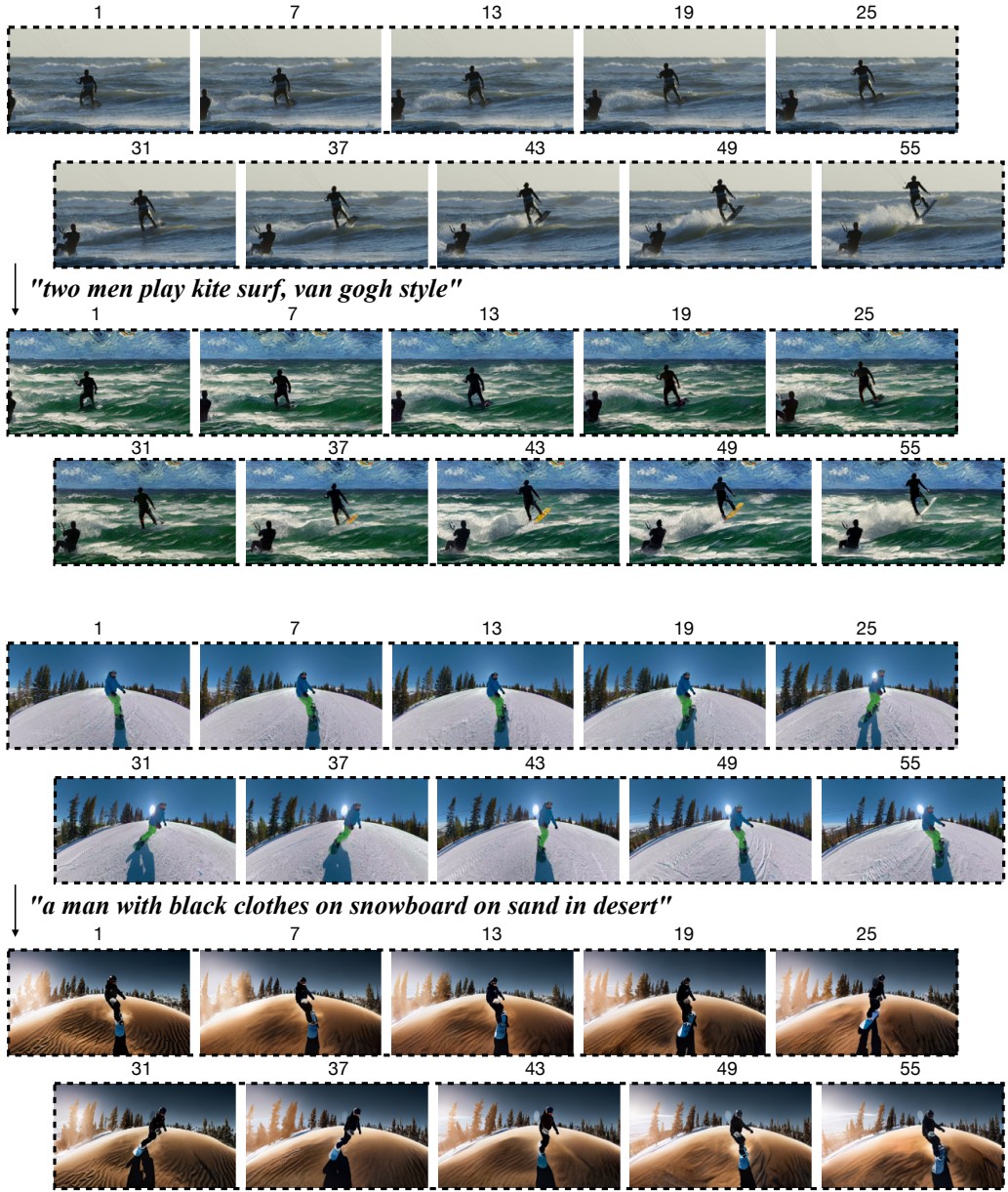

Figure 7: Examples of long video generation. The number indicates the index of frame.

## E    FAILURE CASES

Since we disentangle the video editing tasks into two separate stage, the final performance of the generated video depends on the key frame editing in the first stage. In certain challenging scenarios, the attention-based key frame editing stage struggles to produce consistent frames, primarily due to the complexity of the scene or the presence of exceptionally large motion. In this case, our MaskINT can still interpolate the intermediate frames, albeit with the potential for introducing artifacts. Figure 11 show some failure cases when the first stage fails. In the future work, we are interested in improve the consistency of key frame editing to further improve the performance.

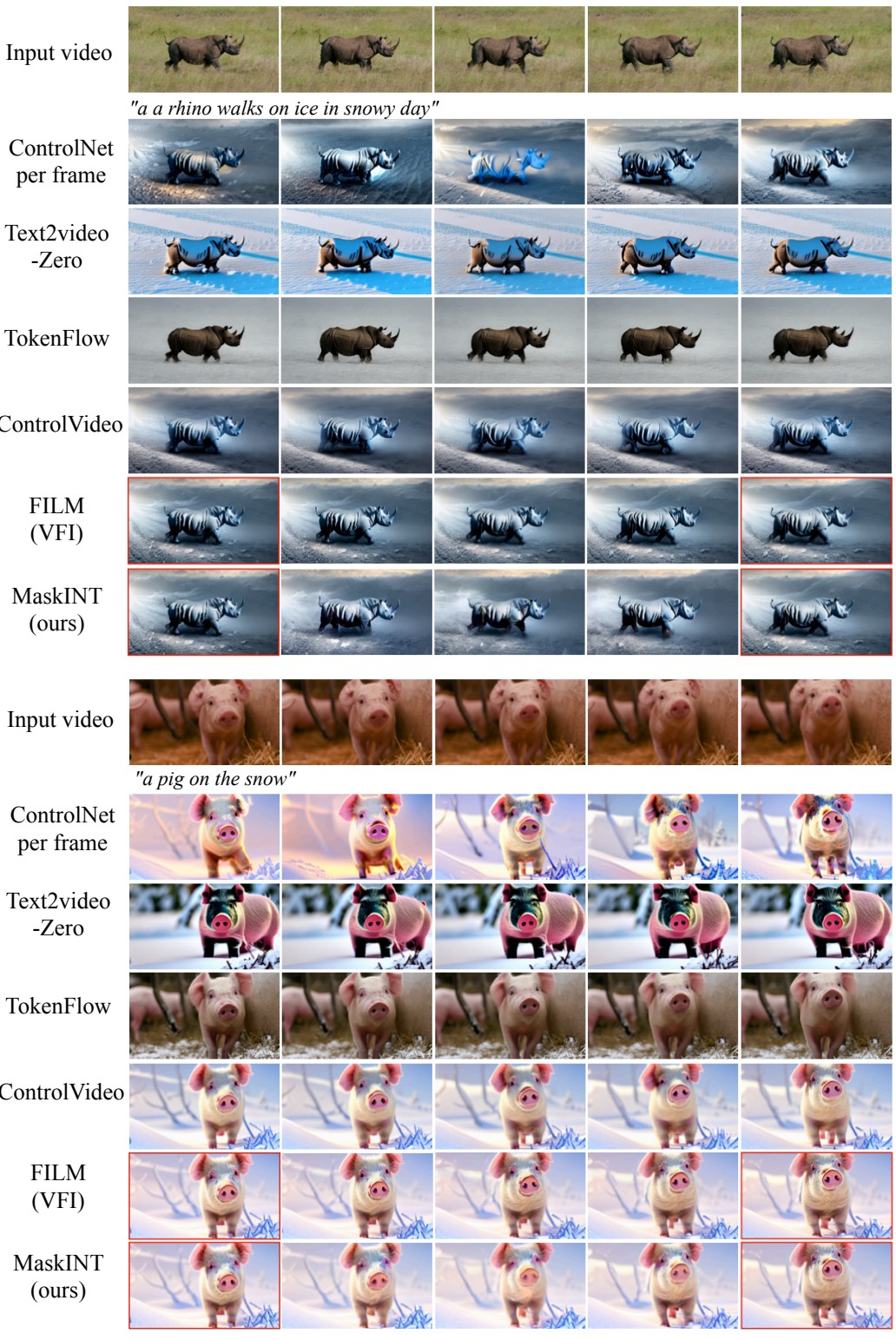

Figure 8: Qualitative comparisons with diffusion-based methods.

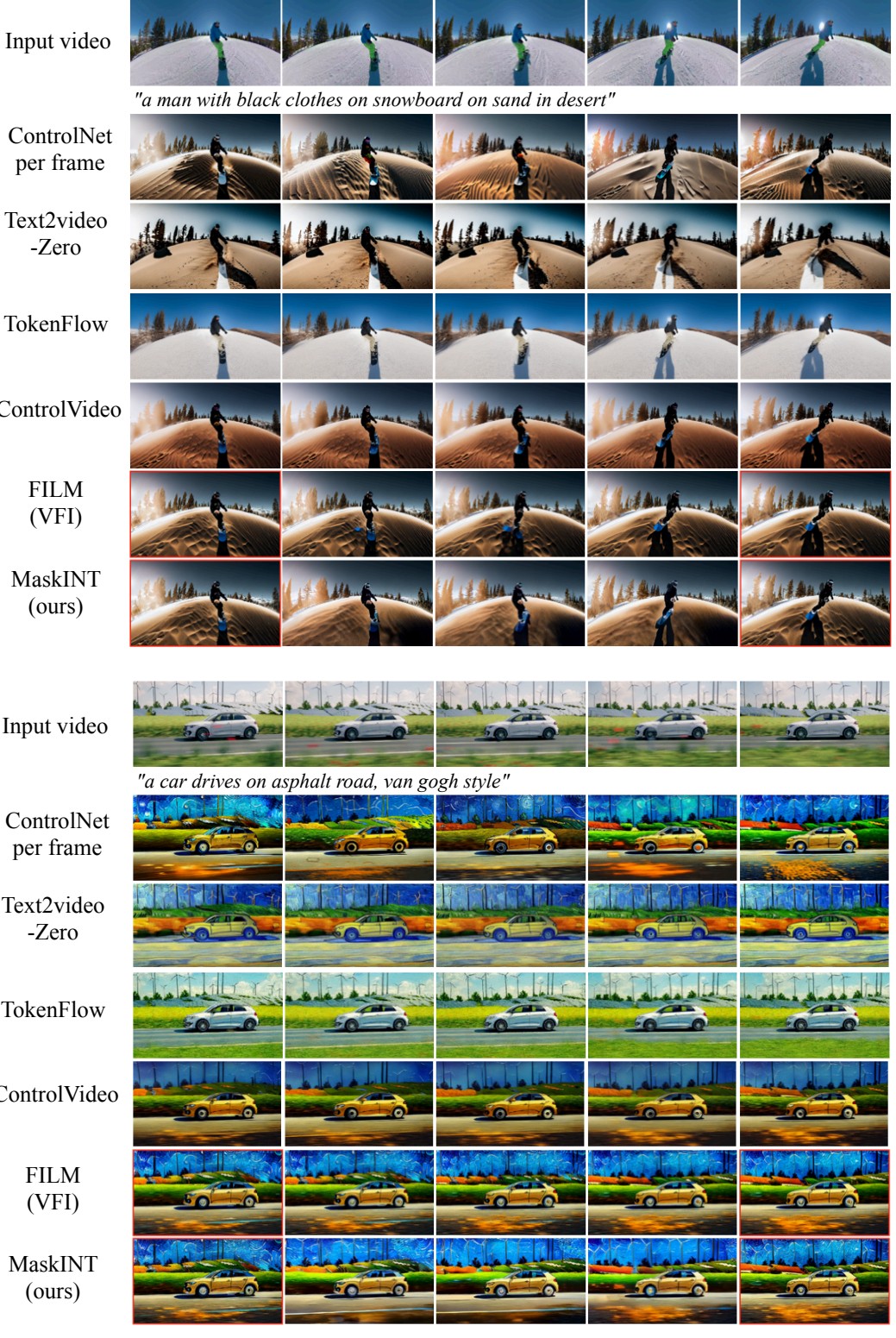

Figure 9: Qualitative comparisons with diffusion-based methods.

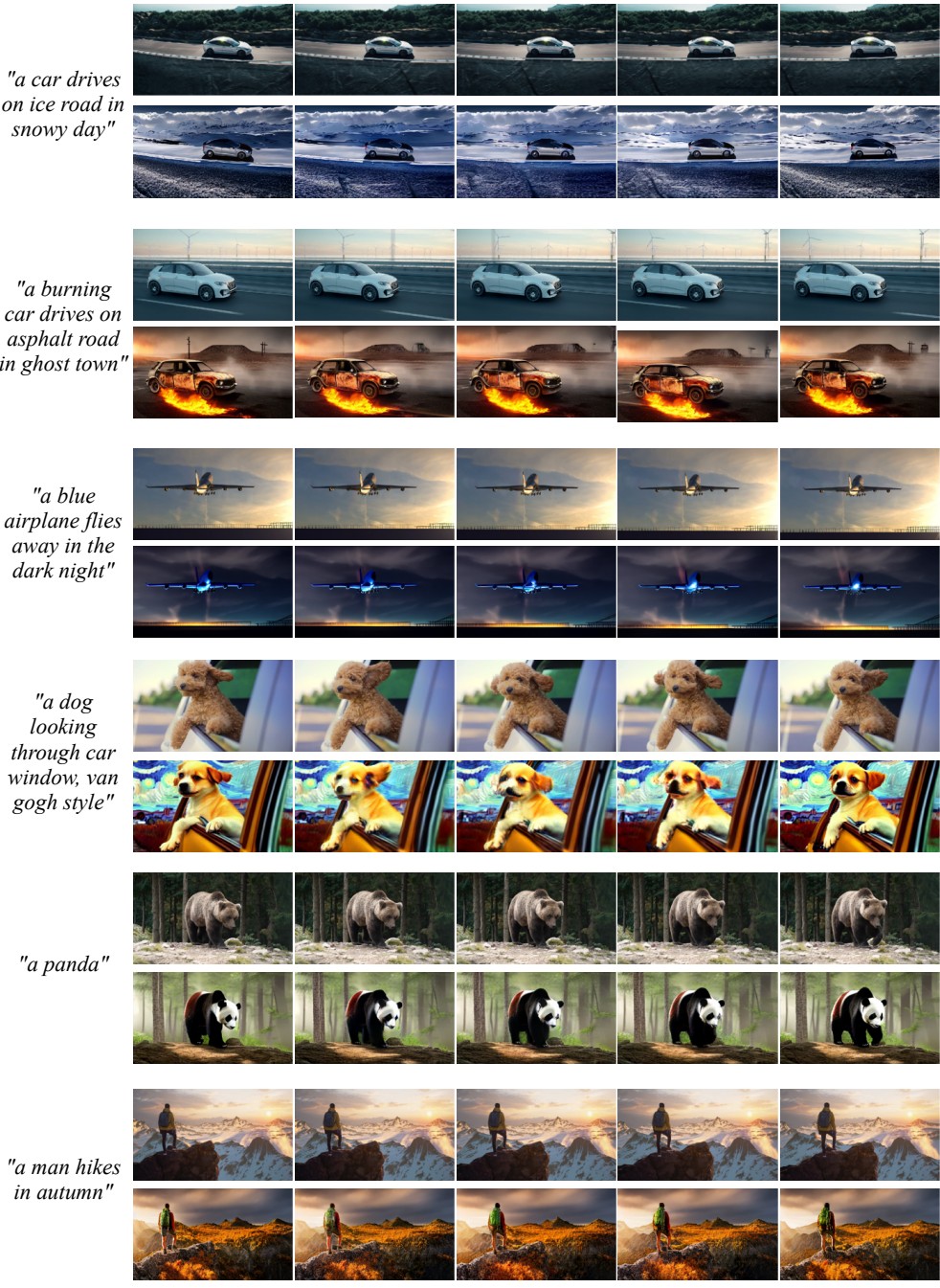

Figure 10: More editing samples with MaskINT.

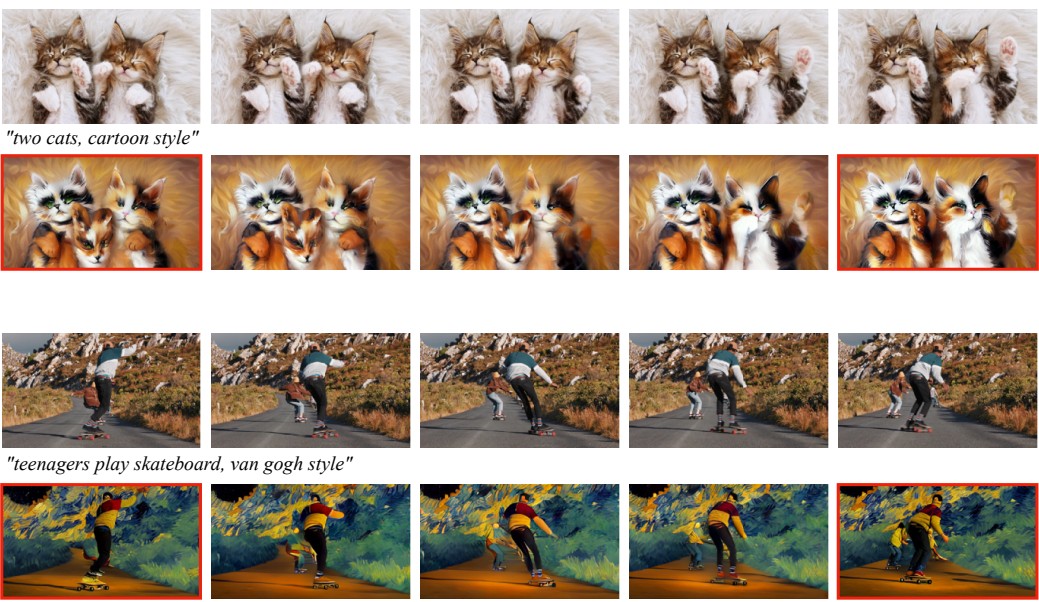

Figure 11: Examples of failure cases.

