# OpenReview forum: "MaskINT: Video Editing via Interpolative Non-autoregressive Masked Transformers"
_ICLR.cc/2024/Conference — ICLR 2024 Conference Withdrawn Submission_

### Official Review · Reviewer_Np2s · 2023-10-23

**Soundness:** 3 good
**Presentation:** 3 good
**Contribution:** 2 fair
**Rating:** 5
**Confidence:** 3

**Summary:**

This paper proposed a two-stage video editing method, which first leverages an off-the-shelf image editing method to edit keyframes, and then performs interpolation between the edited frames. Quantitative and qualitative experiments demonstrate that MaskINT achieves comparable performance with previous methods.

**Strengths:**

- Efficiency. The proposed method achieves comparable performance with diffusion methods, but it is much faster.

**Weaknesses:**

- The video editing performance heavily relies on frame interpolation performance. Almost all showed results (in main submission and Supp) are simple motions, such as car translation, rhino translation. The simple motions can be easily interpolated. But for complex motions, it is difficult to perform frame interpolation, and it also suffer occlusions. Actually, in the showed man dancing case, there are obvious artifacts in arms. Also, the proposed method may suffer a lot in case of long-range video editing. Thus, the generalization ability of the proposed method is somehow limited.
- Evaluation. There are only 11 examples in Supp, and it is difficult to judge the performance. Are the results cherry-picked? Could you give more results?

**Questions:**

- How about the failure cases?
- How about long videos and complex motions?
- Would it fail if the first-stage kerframe editing fails?

---

> ### Author Response · Authors · 2023-11-17
> **Response to Reviewer Np2s**
>
> 1 Generalization ability
> We thank Reviewer Np2s for pointing out the insufficient examples issues. Following your valuable suggestions, we further add more editing samples with diverse objects and scenes in Figure 10 to demonstrate the generalization ability of our method.
>
> 2 Simple motions
> Although the motions such as car translation and rhino translation seem “simple”, they are not easily interpolated with existing video frame interpolation methods. As shown in Table 3 and Figure 6 and also the mp4 videos in Supplementary, current state-of-the-art video frame interpolation methods still fail in many simple motions like raising the head of a camel, the fish movement, etc. The major reason is that these works mainly focus on slow-motion generation, making them less effective in handling frames with large motions. On the contrary, we are pioneering work that proposes the idea of “structure-aware frame interpolation”. With the structure guidance, we can successfully reconstruct the intermediate frames with original motions. It’s true that under complex motions such as the man dancing cases, our method may generate some artifacts. However, we’d like to politely point out that our work mainly focuses on improving the efficiency with masked generative transformers, rather than achieving new state-of-the-art performance. Besides, the current state-of-the-art methods like TokenFlow still generate some artifacts on foot in this challenging case with even 7x inference time.
>
>
> 3 Long video editing
> Since the non-autoregressive pipeline generates all video frames simultaneously, it's challenging for it to edit an entire long video due to GPU memory limitation. Nevertheless, our framework can still be extended to generate long videos by dividing the long video into short clips and progressively performing frame interpolation within each clip. For example, given a video with 60 frames, we can select the 1st, 16th, 31st, 46th, and 60th frames as the key frames. We joint edit these five key frames together and then perform structure-aware frame interpolation within each pair of consecutive key frames. As shown in Figure 7 of Appendix, our method can still successfully generate consistent long videos with this design. The .mp4 video of these generated long videos can be found in Supplementary.
>
> 4 Failure cases
> We also add some failure cases in Figure 11. Since we disentangle the video editing tasks into two separate stages, the final performance of the generated video depends on the keyframe editing in the first stage. In certain challenging scenarios, the attention-based key frame editing stage struggles to produce consistent frames, primarily due to the complexity of the scene or the presence of exceptionally large motion. In this case, our MaskINT can still interpolate the intermediate frames, albeit with the potential for introducing artifacts.

---

### Official Review · Reviewer_hZt7 · 2023-10-31

**Soundness:** 3 good
**Presentation:** 4 excellent
**Contribution:** 3 good
**Rating:** 5
**Confidence:** 4

**Summary:**

The paper proposed a two-stage video editing framework, using T2I diffusion model to edit the key frames and then interpolating between those frames. During T2I diffusion process, the paper leveraged controlnet to jointly keep the edge consistency. After that, a Masked generative transformer model called MaskINT is introduced to generate middle frames. The results show that the proposed network can accelerate generate videos compared with baseline pipelines while suffering slightly temporal and prompt consistency decrease.

**Strengths:**

1. The proposed MaskINT leverage masked generative transformer to interpolate between keyframes.
2. The inference speed outperformed the proposed video editing pipelines.
3. MaskINT is trained on unlabeled video datasets using masked token modeling, without needing text-video pairs.

**Weaknesses:**

1. Although the proposed MaskINT can beat other methods in speed, the method still suffers consistency degradation in both prompt and temporal domain.
2. Noticeable degradation across key frames and interpolated frames.
3. No related baseline comparison between video interpolation pipeline.

**Questions:**

1. By increasing the decoding step and keyframes, the method can increase the performance in Tem-Con and Pro-Con. Can the method reach comparable qualitative results in less time by increasing those hyper parameters?
2. The videos in supplementary seem to have heavy moiré patterns. Why does this occur?

---

> ### Author Response · Authors · 2023-11-17
> **Response to Reviewer hZt7**
>
> 1 Comparison with video interpolation baseline
> Thank you for your suggestions. We further add quantitative comparisons in Figure 5, 8, and 9 with FILM, which is the  state-of-the-art video frame interpolation method. Noticeably, these frame interpolation works cannot intermediate frames following the original motions, due to the lack of structural motions. Besides, we also quantitatively compare our method with FILM and RIFE under the “reconstruction” setting suggested by Reviewer gZ41. The results in Table 3 and Figure 6 also demonstrate the benefit of our method.
>
>
> 2 Degradation in performance
> We’d like to politely emphasize that our work does not aim to achieve state-of-the-art performance, but would rather focus more on the efficiency area with a better trade-off between performance and efficiency.
>
> 3 Ablation study on the number of key frames and decoding steps
> As shown in Table 2, when further increasing the number of key frames, we can further get improvement because a better understanding of motion can be extracted from more frames. However, when further increasing the number of decoding steps, it tends to reach a saturation point. This is consistent with the conclusion in MaskGiT.
>
>
> 4 Moiré patterns in supplementary videos
> Thanks for pointing out this issue in the quality of video visualization. The reason mainly comes from the “.gif” format. We save them into the ".mp4” file, and this problem can be solved.

---

### Official Review · Reviewer_gZ41 · 2023-11-01

**Soundness:** 3 good
**Presentation:** 3 good
**Contribution:** 2 fair
**Rating:** 6
**Confidence:** 4

**Summary:**

This paper introduces a new approach to structure-guided editing for videos. The proposed method is composed of two stages. In the first stage, an image-based diffusion model is leveraged, along with the cross-frame attention technique, to jointly edit a small number of key frames. In the second stage, a structure-guided non-autoregressive masked transformers model is developed for the interpolation task, aiming to propagate the information from the (edited) key frames to the intermediate frames. The experiments in the paper demonstrate the proposed method can enable temporally consistent edit propagation results while achieving better efficiency compared to existing diffusion-based approaches.

**Strengths:**

The video editing results provided in the paper demonstrate that the non-autoregressive masked generative modelling technique, which have mostly been applied to the unconditional generation or text-condition generation so far, can be effectively adapted to the structure-conditioning generation setting.

The experiments in the paper demonstrate that the proposed method can achieve better efficiency compared to existing diffusion-based approaches.

**Weaknesses:**

While the idea of extending the non-autoregressive masked transformer technique to structure-guided generation is technically sound, the technical contribution on the fundamental side is somewhat limited. Video editing with diffusion model via key-frame edit propagation has been widely explored. The effectiveness of masked generative model in video generation has also been well established. The key contribution of this paper, from my perspective, is in showing that it is possible to incorporate dense structure information into the masked transformer model. There are limited discussions in the paper, however, to provide insights on why such a task is difficult, what are the fundamental challenges in doing that, and why the proposed technique is a good solution for such challenges.

The discussion on the technical details is somewhat vague. In particular, it seems that the model architecture details were not elaborated.

The provided evaluation is a bit weak:
+ I feel that the subjective comparison should be made more complete: video results were only provided (in the supplementary material) for the proposed method, not for competing methods. That makes it difficult to assess the temporal quality of the proposed method in comparison with the other methods.
+ The comparison is not entirely fair, competing methods are all zero-shot setup, which never trains a video model. Existing video diffusion works have been shown to be effective for interpolation ([1], [2]), a fair comparison would be to compare with adapted versions of those methods to incorporate structure control signal.
+ It seems that the provided results are all with stylized content, which tends to make it more visually tolerable to temporal inconsistencies. As the main goal of the second-stage model is to perform keyframe propagation, I think one important test that should be done is to apply the model on the reconstructive setting, i.e. perform propagation with the original keyframes instead of edited ones and assess the reconstruction quality of the intermediate frames.

[1] Make-A-Video: Text-to-Video Generation without Text-Video Data. Singer et al., 2022

[2] Align your Latents: High-Resolution Video Synthesis with Latent Diffusion Models. Blattmann et al., 2023

**Questions:**

Please find my detailed comments in the Weaknesses section. Other than that, there are a couple of questions I’m curious about:
+ Will the proposed technique works for other type of controls such as depth maps or pose map?
+ How will the method perform in the extrapolation instead of the interpolation setting? Or in the setting where only one key frame is edited?

---

> ### Author Response · Authors · 2023-11-17
> **Response to Review gZ41**
>
> 1 Results on Video Frame Interpolation with original key frames.
> Thanks for pointing out this important evaluation setting. Following your valuable suggestions, we conduct a quantitative and qualitative evaluation of the performance of MaskINT in the reconstructive setting, where it engages in video frame interpolation using the original key frames. In this evaluation, we apply signal-to-noise ratio (PSNR), learned perceptual image patch similarity (LPIPS), and structured similarity (SSIM) to compare the interpolated frames with original video frames. We benchmark our method against two state-of-the-art Video Frame Interpolation (VFI) methods, including FILM [A] and RIFE [B]. Table 3 and Figure 6 in Appendix show that our method significantly outperforms VFI methods on all evaluation metrics, with the benefit of the structure guidance from the intermediate frames (For better visualization, we suggest you watch the .mp4 video in Supplementary).
> Moreover, even when confronted with significant motion between two frames, our approach successfully reconstructs the original video, maintaining consistent motion through the aid of structural guidance. In contrast, FILM introduces undesirable artifacts, including distorted background, multiple cat hands, and the absence of a camel's head, etc. The major reason is that current VFI models mainly focus on generating slow-motion effects and enhancing frame rate, making them less effective in handling frames with large motions. Additionally, the absence of structural guidance poses a challenge for these methods in accurately aligning generated videos with the original motion.
>
>
> 2 Comparisons with [1] and [2] with additional structure control.
> Thanks for sharing these important works. However, We would like to respectfully point out that designing an architecture to explicitly introduce structural control to these works is a non-trivial task.Besides, these methods are not open-source and they use some in-house datasets to train the model, making it difficult to make any further adjustments and conduct a fair comparison.
>
>
> 3 Performance with single frame.
> We further evaluate the performance where only the initial edited key frame is given. As shown in Table 2, when there is only one keyframe, the performance is downgraded since it’s difficult to understand the motion within the video with a single reference frame.
>
>
>
> [A] Reda, Fitsum, et al. "Film: Frame interpolation for large motion." European Conference on Computer Vision. Cham: Springer Nature Switzerland, 2022.
>
> [B] Huang, Zhewei, et al. "Real-time intermediate flow estimation for video frame interpolation." European Conference on Computer Vision. Cham: Springer Nature Switzerland, 2022

---

### Author Response · Authors · 2023-11-17
**Thanks for all reviewers and Summary of changes in the revision**

We sincerely thank all reviewers for insightful comments and constructive feedback on our paper. To address the concerns raised by the reviewers, we have updated our manuscript. Here we summarize the changes in the revision:

1. Following the suggestions from all reviewers, We add quantitative and qualitative comparisons with state-of-the-art Video Frame Interpolation (VFI) methods with both original key frames and edited key frames.

2. Following the suggestions from Reviewer Np2s, we add more editing samples and some failure cases to demonstrate the generalization of our method.

3. Following the suggestions from Reviewer Np2s, we extend our framework to long video editing.

4. Following the suggestions from Reviewer hZt7, we add ablation studies of increasing the number of edited key frames.

5. Following the suggestions from Reviewer hZt7, we update the .gif videos in Supplementary Materials to .mp4 format, which help get rid of the moiré pattern.